# Microglia Mitigate Neuronal Activation in a Zebrafish Model of Dravet Syndrome

**DOI:** 10.3390/cells13080684

**Published:** 2024-04-15

**Authors:** Alexandre Brenet, Julie Somkhit, Zsolt Csaba, Sorana Ciura, Edor Kabashi, Constantin Yanicostas, Nadia Soussi-Yanicostas

**Affiliations:** 1NeuroDiderot, INSERM U1141, Université Paris Cité, Robert Debré Hospital, 75019 Paris, Franceconstantin.yanicostas@inserm.fr (C.Y.); 2Institut Imagine, University Paris Descartes, Necker-Enfants Malades Hospital, 75015 Paris, France; 3INSERM, T3S, Department of Biochemistry, Université Paris Cité, 75006 Paris, France

**Keywords:** microglia, epilepsy, seizure, calcium imaging, zebrafish, seizures, cytokines, neuroprotection

## Abstract

It has been known for a long time that epileptic seizures provoke brain neuroinflammation involving the activation of microglial cells. However, the role of these cells in this disease context and the consequences of their inflammatory activation on subsequent neuron network activity remain poorly understood so far. To fill this gap of knowledge and gain a better understanding of the role of microglia in the pathophysiology of epilepsy, we used an established zebrafish Dravet syndrome epilepsy model based on Scn1Lab sodium channel loss-of-function, combined with live microglia and neuronal Ca^2+^ imaging, local field potential (LFP) recording, and genetic microglia ablation. Data showed that microglial cells in *scn1Lab*-deficient larvae experiencing epileptiform seizures displayed morphological and biochemical changes characteristic of M1-like pro-inflammatory activation; i.e., reduced branching, amoeboid-like morphology, and marked increase in the number of microglia expressing pro-inflammatory cytokine Il1β. More importantly, LFP recording, Ca^2+^ imaging, and swimming behavior analysis showed that microglia-depleted *scn1Lab*-KD larvae displayed an increase in epileptiform seizure-like neuron activation when compared to that seen in *scn1Lab*-KD individuals with microglia. These findings strongly suggest that despite microglia activation and the synthesis of pro-inflammatory cytokines, these cells provide neuroprotective activities to epileptic neuronal networks, making these cells a promising therapeutic target in epilepsy.

## 1. Introduction

Epilepsy, a frequent neurological disorder affecting more than 1% of the world population [1,2], is characterized by recurrent seizures caused by the synchronized hyperactivation of neuronal networks of various sizes, ranging from limited brain regions to the whole cortex, leading to focal or generalized seizures, respectively [3,4,5]. Although potent anti-epileptic drugs (AEDs) have been developed in the last decades, they are ineffective in one-third of patients, approximately, and especially children with developmental epileptic encephalopathies (DEE), such as Dravet syndrome (DS) [6,7,8]. Moreover, several AEDs have been shown to display teratogenic activities in pregnant women [9,10]. These limitations emphasize the need for novel AEDs and new anti-epileptic strategies. Notably, while almost all current AEDs are “neurocentric”, targeting neuron proteins directly involved in neuron network activation, microglial cells and their inflammatory activation in epileptic contexts have received little attention so far as therapeutic targets [11,12].

Microglial cells, the resident brain macrophages, display several essential functions, including immuno-protection, maintenance of neuron health and homeostasis, and maturation of neuronal networks through developmentally controlled phagocytosis of supernumerary apoptotic neurons and pruning of excess synapses [13]. Moreover, as immune cells, microglia respond rapidly to brain injuries or infections through complex reprogramming, referred to as microglia activation, which comprises morphological and biochemical changes, including increased motility and phagocytic activities, associated with the synthesis and release of pro-inflammatory cytokines [12].

In epileptic patients, it was shown for a long that epileptic seizures induce microglia activation and the synthesis of inflammatory mediators [11,14,15,16]. Actually, analysis of post-mortem brain tissues from epileptic patients revealed massive microglia activation, directly correlated with seizure severity [17,18]. Moreover, in various pharmacological rodent epilepsy models, significant morphological changes of microglial cells were observed after seizures, prior to the onset of neuronal damages [19,20], suggesting that microglia activation is an early event following neuron seizures.

In epileptic patients and animal models [21], while data suggested that microglia might be harmful to hyperexcitated neuronal networks, probably owing to the synthesis and release of pro-inflammatory cytokines, which are known to promote neuron excitation [22,23,24], other studies suggested that microglia played a beneficial role by decreasing neuronal network hyperactivity [19,25,26,27]. Data also suggested that microglia response to epileptic seizures is not uniform throughout the brain, with microglia showing a wide range of phenotypes within the same individual [18]. Moreover, other data indicated that the synthesis of both pro- and anti-inflammatory cytokines is increased in microglia after epileptic seizures, supporting the notion that the response of these cells to seizures is not limited to classic M1-type pro-inflammatory activation [27,28]. Therefore, whether microglia activity has an overall detrimental or beneficial role in epilepsy is still a largely open question with far-reaching therapeutic implications [26].

To address this issue and gain a better understanding of the role of microglia in epilepsy, we used a zebrafish genetic model of DS, a severe intractable epileptic syndrome in infants, which is caused in 70–80% of the cases by de novo loss-of-function mutations in the gene encoding the voltage-gated sodium channel, *SCN1A* [29,30,31]. Using a well-characterized zebrafish DS model relying on the loss of function of the zebrafish orthologue of *SCN1A*, the *scn1Lab* gene, we first investigated the morphological changes of microglial cells in vivo using the transgenic Tg[mpeg1:mCherryF] microglia-reporter line. We also recorded neuron activity using neuronal Ca^2+^ imaging and local field potential (LFP) recording in *scn1Lab*-deficient larvae, with or without microglia.

Our data showed that microglia are deeply remodeled in *scn1Lab*-deficient larvae, with cells shifted toward an amoeboid morphology with markedly reduced branching, combined with significantly increased mobility and number of Il1β-expressing cells. In addition, in *scn1Lab* larvae fully devoid of microglia, neuronal hyperactivation was significantly increased when compared to that observed in *scn1Lab* individuals not microglia-depleted, suggesting these cells have neuroprotective activities for epileptic neurons that decrease their hyperactivity.

## 2. Materials and Methods

### 2.1. Fish Maintenance and Transgenic Lines

Zebrafish were maintained at 26.5 °C in 14 h light and 10 h dark cycles. Embryos were collected by natural spawning and raised at 28 °C in an E3 medium. Developmental stages were determined as hours post-fertilization (hpf) or days post-fertilization (dpf), as previously described. 0.003% 1-phenyl-2-thiourea (PTU) was added at 1 dpf to avoid pigmentation. The *scn1Lab^s552^* line has been previously described [30] and was generously provided by the laboratory of Dr. Herwig Baier. Calcium imaging was performed using the Tg[Huc: GCaMP5G] transgenic line [32,33], and microglia and cytokine Il1β expression were visualized using the transgenic Tg[mpeg1: mCherryF] [34] and Tg[il1β: GFP] lines, respectively [35,36].

All animal experiments described in this study were conducted at the French National Institute of Health and Medical Research (INSERM), UMR 1141, Paris, in compliance with European Union guidelines for the handling of laboratory animals (https://environment.ec.europa.eu/topics/chemicals/animals-science_en, accessed on 10 April 2024), and were approved by the Direction Départementale de la Protection des Populations de Paris and the French Animal Ethics Committee under reference No. 2012-15/676-0069.

### 2.2. Morpholino Knockdown

The following morpholinos (MOs), obtained from Gene Tools (Philomath, OR, USA), were used in this study: MO 5′-CTGAGCAGCCATATTGACATCCTGC-3′ was used at 0.62 mM to block zebrafish *scn1Lab* mRNA transcription, and MO 5′-GATATACTGATACTCCATTGGTGGT-3′ was used at 0.88 mM to block zebrafish *pu.1* mRNA transcription. These MOs were injected together with 0.03 mM rhodamine B dextran and 0.1 mM KCl into 1–2 cell stage embryos.

### 2.3. Microglia Morphological Analysis

To visualize microglial cells in *scn1Lab* embryos experiencing epileptiform seizures, 4 dpf Tg[mpeg1:mCherryF]; *scn1Lab* morphants, and Tg[mpeg1:mCherryF] controls, were paralyzed using 300 µM pancuronium bromide (Sigma Aldrich, Saint-Louis, MO, USA) and agar-mounted in the center of a 35 mm glass-bottom dish. 100 µm stacks of the larval optic tectum were acquired at 1024 × 1024-pixel resolution using a Leica SP8 laser scanning confocal microscope equipped with a 20×/0.75 multi-immersion objective. Images were processed using AutoQuant X3 (Media Cybernetic, Rockville, MD, USA) and Fiji ImageJ 1.53 (NIH, Bethesda, MD, USA) softwares. Microglia surface area, volume, and sphericity were quantified using Imaris MeasurementPro 9.1 (Oxford Instruments, Abingdon, UK). The number and length of microglia processes and Sholl analysis were determined using the Imaris Filament Tracer 9.1 (Oxford Instruments, Abingdon, UK).

### 2.4. Morphological Clustering of Microglial Cells

Cluster analysis, based on microglia morphology parameters, was performed as described in [37]. Briefly, five morphological parameters were quantified for each microglial cell: sphericity (S), number of processes (NP), total process length (TL), surface area (A), and volume (V). Cells with at least three of the following parameters: sphericity < 0.5, process number > 7, total process length > 140 µm, and surface area > 2400 µm^2^, were classified as “branched”. Conversely, microglia with at least three of the following parameters: sphericity > 0.7, process number < 3, total process length < 60 µm, and surface area < 1500 µm^2^, were classified as “amoeboid”. Lastly, cells showing less than three “branched” or “amoeboid” parameters were classified as “transitional” microglia.

### 2.5. RNA Isolation and Quantitative RT-PCR

For RNA isolation, whole larvae or dissected brains were homogenized using a syringe equipped with a 26G needle (10 embryos or brains per sample) using the RNA XS Plus kit (Macherey Nagel, Düren, Germany). Following RNA quantification with a Nanodrop 2000 (ThermoScientific, Waltham, MA, USA) and RNA integrity analysis using denaturing gel electrophoresis, total RNAs (1 µg) were reverse transcribed into cDNAs using the iScript cDNA Synthesis Kit (Bio-Rad, Hercules, CA, USA), and qPCRs were performed using the iQ SYBR Green Supermix (Bio-Rad, Hercules, CA, USA). Samples were run in triplicate, and the expression levels of the studied genes were normalized to those of the tbp gene. The primers (Eurofins Genomics, Ebersberg, Germany) used are listed in Appendix A.

### 2.6. Analysis of the Dynamics of Microglia Morphology Changes

Here, 4 dpf pancuronium bromide-paralyzed Tg[mpeg1:mCherryF] *scn1Lab* morphant and not-injected Tg[mpeg1:mCherryF] control larvae were agar-mounted in the center of a 35-mm glass-bottom dish. A 42-µm stack of the larval living half optic tectum was acquired at 1024 × 512-pixel resolution every 40 s for 1 h using a Leica SP8 laser scanning confocal microscope equipped with 40×/1.1 water objective (Leica, Wetzlar, Germany). Videos were processed using ImageJ 1.53 software. Microglial cell body displacement, distance traveled between the first and last frames and microglial cell body speed displacement were quantified using Imaris MeasurementPro 9.2.

### 2.7. Calcium Imaging

Quantification of neuronal calcium transients was performed as previously described [31,37]. Briefly, 4 dpf Tg[Huc:GCaMP5G] *scn1Lab* morphant and Tg[mpeg1:mCherryF] control larvae were paralyzed using pancuronium bromide, agar-mounted in the center of a recording chamber, and placed under a Leica SP8 laser scanning confocal microscope equipped with a 20×/0.75 multi-immersion objective. A single focal plane of the optic tectum was captured at a high frequency for 40 min. The mean gray value of each frame was measured using ImageJ software, and fluorescence variation (ΔF/F_0_) was calculated using Excel (Microsoft, Redmond, WA, USA) by subtracting the mean fluorescence intensity of all frames from the fluorescence intensity of a frame of interest, then normalizing this difference by the mean fluorescence intensity of all frames. The fluorescence drift during recording was corrected by subtracting the background intensity of the surrounding frame. Calcium events were defined as any fluorescence increase greater than 0.04 ΔF/F_0_.

For movie illustration (Appendix A), larvae were paralyzed and immobilized as described above. Calcium transients in a single focal plane of the optic tectum of larvae were captured for 5 min at 5 Hz using a Yokogawa CSU-X1 spinning disk confocal microscope (Yokogawa, Tokyo, Japan) equipped with an LD LCI Plan-Apochromat 25×/0.8 DIC Imm Korr (UV) objective (Zeiss, Oberkochen, Germany).

### 2.8. Local Field Potential (LFP) Recording

LFP recording was performed as previously described [31]. Briefly, 4 dpf *scn1Lab* morphant or not-injected control larvae were paralyzed using pancuronium bromide and agar-mounted in the center of a recording chamber. A glass electrode filled with artificial cerebrospinal fluid was placed in the right neuropil of the optic tectum. Neuronal activity was recorded for 1 h in current clamp mode at a 10 µs sampling interval, amplified with a digital gain of 10, and filtered through a 0.1 Hz high-pass filter and a 1 kHz low-pass filter. Recordings were analyzed using Clampfit 11 software (Molecular Devices, San Jose, CA, USA). Every downward membrane potential variation under −0.3 mV amplitude and lasting more than 100 ms was defined as an event.

### 2.9. Locomotor Activity

The locomotor behavior of larvae was assessed using an infrared automated tracking device (Zebrabox), controlled by the ZebraLab 5.31 software (ViewPoint, Lyon, France). First, 4 dpf larvae were individually deposited in 96-well plates in 200 µL of E3 medium. The plates were then placed in the Zebrabox recording chamber in the dark for 30 min for habituation. Then larvae were tracked for 30 min divided into one-minute integration time. The animal color was set to dark and the detection sensitivity to 12 for tracking. Besides the total distance swum by the larvae, we also recorded the time during which larvae were either immobile or in low-, medium-, or high-speed movements. The inactive speed threshold was set at 4 mm and the high-speed threshold was set at 8 mm. Each experiment replicates comprised at least 16 larvae per condition and was replicated at least three times.

### 2.10. Data Analysis

Data were analyzed and plotted using R-studio 1.2.5. (Posit, Boston, MA, USA). Data are presented as mean ± sem. Statistical analysis was performed using the Mann–Whitney test if the data did not follow a normal distribution or the Student’s unpaired *t*-test with or without Welch’s correction depending on the variance difference. A two-way ANOVA with a Tukey post-test was used to analyze the data in Figures 3–5.

## 3. Results

### 3.1. Microglia Morphology Changes in scn1Lab-KD Larvae

To characterize microglia morphology changes in epileptic brains in vivo, we first took advantage of real-time microglia imaging in living zebrafish embryos [36], using two zebrafish *scn1Lab* models of DS, *scn1Lab* morphant [31] and *scn1Lab^s552/s552^* mutant larvae [38]. Both models display recurrent spontaneous epileptiform seizures starting from 4 days post-fertilization (dpf) onward, as visualized by LFP recording and Ca^2+^ imaging and by the characteristic swirling behavior of larvae [31,38,39]. Although, throughout this study, we used *scn1Lab* morphants, referred to below as *scn1Lab*-KD individuals, all data were confirmed in *scn1Lab^s552/s552^* mutant larvae, displaying a complete loss of Scn1Lab function [31,38]. To visualize microglial cells in vivo, we chose the transgenic Tg[mpeg1:mCherryF] line, which shows intense labeling of these cells [40]. However, as the *mpeg1:mCherryF* reporter gene is expressed in both microglia and macrophages, we verified that all the mCherryF-expressing microglia imaged in this study also expressed the Tg[p2y12:GFP] transgene [41], a specific marker of microglial cells [42].

As previously shown [36,43], in wild-type (WT) larvae, most microglial cells displayed the characteristic branched phenotype of “resting” microglia, with a relatively small cell body and several long and ramified processes (Figure 1(A_1_–A_3_) and Appendix A (Figure 1A_2_): (Accessed on 10 April 2024, Link: https://urlz.fr/iPxZ). By contrast, in *scn1Lab*-KD larvae, a significantly increased number of microglia exhibiting a more rounded morphology, a larger cell body, and much fewer and shorter branches were observed (Figure 1(B_1_–B_3_) and Appendix A (Figure 1B_3_): (Accessed on 10 April 2024, Link: https://urlz.fr/iPxP). To further characterize and quantify these morphological differences, we used the 3D image analysis software Imaris 9.2 (Bitplane) to measure several microglia morphological parameters, as described in [44] (sphericity, volume, surface area, branch number, total branch length, and ramification index). Data confirmed that microglia in *scn1Lab*-KD larvae displayed increased sphericity (WT: 0.567 ± 0.006 vs. *scn1Lab*-KD: 0.644 ± 0.006; *p* < 0.0001) (Figure 1C), a decreased number of branches (WT: 6.3 ± 0.2 vs. *scn1Lab*-KD: 5.1 ± 0.2; *p* < 0.0001) (Figure 1D), a decreased total (WT: 115 ± 3 µm vs. *scn1Lab*-KD: 85 ± 3 µm; *p* < 0.0001) (Figure 1E) and mean process lengths (control: 23.5 ± 0.4 µm vs. *scn1Lab*-KD: 19.6 ± 0.3 µm; *p* < 0.0001) (Figure 1F), and a decreased ramification index (WT: 1.94 ± 0.05 vs. *scn1Lab*-KD: 1.74 ± 0.05; *p* < 0.0001) (Figure 1G), when compared to those observed for microglia in WT individuals. Sholl analysis, a method used to quantify both the extent and complexity of cell branches (Imaris 9.2 software), further confirmed that microglia in *scn1Lab*-KD individuals had fewer and shorter branches than those in their WT siblings (Figure 1H). Taken together, our data show that microglia exhibited a morphology reminiscent of M1-type activated macrophages in *scn1Lab*-KD larvae.

To further evaluate the extent of microglia morphology changes in *scn1Lab*-KD larvae compared to that seen in their non-injected sibling controls, we performed a cluster analysis of these cells according to five morphological parameters (sphericity, branch number, branch length, surface area, and volume), leading to three cell clusters, as described in [37] (see Materials and Methods). The first comprised low-sphericity microglia with numerous elongated branches and high surface area and volume, which were referred to below as “branched” microglia. The second included cells showing high sphericity, few short processes, and low surface area and volume, which were referred to below as “amoeboid” cells. The third, which comprised cells displaying intermediate phenotypes, was referred to below as “transitional” microglia (Figure 1J). Although the three microglia clusters were found in both contexts, their respective proportions varied greatly. In *scn1Lab*-KD larvae, we observed a high percentage of “amoeboid” microglia (29.1 ± 1.2% vs. 10.4 ± 0.3%; *p* < 0.001) combined with a low proportion of “branched” cells (10.2 ± 0.8% vs. 27.8 ± 1.5%; *p* < 0.01) (Figure 1K), compared to that seen in WT individuals. However, the percentage of cells showing an “intermediate” phenotype was roughly similar in *scn1Lab*-KD and control larvae (61.8 ± 1.8% vs. 61.8 ± 1.9%; *p* > 0.05). These findings further suggest that a significantly increased number of microglia display an M1-like inflammatory phenotype in *scn1Lab*-KD larvae.

Visual examination of real-time recording of the behavior of microglia in *scn1Lab*-KD and WT larvae provided a first indication that these cells, in addition to their morphological changes, displayed increased mobility in the mutant context. To confirm this observation, we measured the movements of microglia cell bodies and their speed in the two genetic contexts. In WT larvae, as previously described [39], microglia displayed several elongated branches that were constantly extending and retracting, while their cell bodies remained almost immobile (Figure 2(A_1_,A_2_,C_1_,F_1_)) (Appendix A, (Accessed on 10 April 2024, Link: https://urlz.fr/iOIz). By contrast, microglia in *scn1Lab*-KD larvae were much less branched, but their cell bodies were significantly more mobile (Figure 2(B_1_,B_2_,C_2_,F_2_)) (Appendix A, (Accessed on 10 April 2024, Link: https://urlz.fr/iOIA).

### 3.2. Increased Microglia-Mediated Brain Inflammation in scn1Lab-KD Larvae

Because the microglia phenotype changes observed in *scn1Lab*-KD larvae were strongly reminiscent of those of “activated” M1-type inflammatory microglia observed in various neuronal disease situations, including epilepsy [18], we next evaluated the neuroinflammatory profile in the brain of *scn1Lab*-KD and WT larvae. First, we used qRT-PCR to quantify the brain accumulation of transcripts encoding two major pro- (Il1β and Il8) and three anti-inflammatory cytokines (Il10, Il4, and Tgfβ3). Surprisingly, we did not observe any significant changes, neither in the expression of pro-[*il1β* (WT: 1.00 ± 0.22 vs. *scn1Lab*-KD: 1.33 ± 0.18; *p* > 0.05) and *il8* (WT: 1.00 ± 0.09 vs. *scn1Lab*-KD: 0.93 ± 0.11; *p* > 0.05)] (Figure 3A), nor anti-inflammatory cytokine mRNAs [*il10* (WT: 1.00 ± 0.21 vs. *scn1Lab*-KD: 0.767 ± 0.17; *p* > 0.05), *il4* (WT: 1.00 ± 0.12 vs. *scn1Lab*-KD: 0.89 ± 0.19; *p* > 0.05), and *tgfβ3* (WT: 1.00 ± 0.04 vs. *scn1Lab*-KD: 0.876 ± 0.08; *p* > 0.05)] (Figure 3B). Therefore, although morphological changes seen in *scn1Lab*-KD larvae suggested pro-inflammatory microglia activation, no significant increase in the expression of transcripts encoding pro- or anti-inflammatory cytokines could be detected in the brains of these individuals. It can be hypothesized that levels of neuroinflammation in this genetic epilepsy model are much lower than those observed in rodent pharmacological models, as also suggested by the relatively small percentage of “amoeboid” microglia in *scn1Lab*-KD individuals (~30%). To analyze more precisely microglia expression of *il1β* mRNA, encoding a major pro-inflammatory cytokine, we generated Tg[mpeg1:mCherryF]; *scn1Lab*-KD larvae also carrying the Il1β-reporter Tg[*il1β*:GFP] transgene [45]. Data revealed a significant increase in the number of Il1β-expressing microglia in *scn1Lab*-KD larvae when compared to that observed in not-injected controls (WT: 14.6 ± 1.7% vs. *scn1Lab*-KD: 33.6 ± 1.7%; *p* < 0.0001) (Figure 3C–F). Interestingly, as can be seen in Figure 3D_2_, while part of the “amoeboid” microglial cells only expressed the *il1β:GFP* transgene in *scn1Lab*-KD larvae, few highly branched cells also expressed the transgene, highlighting the complexity of the microglia response. Altogether, these results showed a significant increase in the percentage of activated il1β-expressing microglia in the mutant context, consistent with increased microglia activation and brain neuro-inflammation.

### 3.3. Microglia Ablation Increases Neuronal Activation in scn1Lab-KD Larvae

The phenotype changes and increased Il1β expression observed in microglia of *scn1Lab*-KD larvae further suggest that a significant proportion of microglia, at least 30% of the cells, respond to neuron hyperactivity through M1-type activation. These results then raised the question as to whether this inflammatory response of microglia mitigates or worsens the excitability of neuronal networks caused by Scn1lab loss-of-function. As a first attempt to address this question, we generated *scn1Lab*-KD larvae fully devoid of microglia following morpholino-mediated inactivation of the *pu.1* gene, encoding a factor essential for proper differentiation of macrophages and microglia [35,46,47], and also expressing the Tg[HuC:GCaMP5G] transgene, a sensitive calcium bio-sensor [47]. As previously described [43], the brains of *pu.1* morphant larvae are completely devoid of microglial cells, as shown by the absence of L-plastin immunostaining (Figure 4A_1_,A_2_). By taking advantage of these morphant individuals without microglia, we were able to compare neuron activation in WT and *scn1Lab*-KD larvae, with or without microglia, using simultaneous calcium imaging and local field potential (LFP) recording.

As previously shown [31,47], when compared to WT individuals, *scn1Lab*-KD larvae display a markedly increased neuronal activity, as indicated by the larger number of seizure-like events revealed by both calcium imaging (WT: 16.5 ± 1.0 vs. *scn1Lab*-KD: 32.2 ± 1.4, *p* < 0.0001) (Figure 4B,D_1_) (Appendix A, (Accessed on 10 April 2024, Link: https://urlz.fr/jlSX), (Appendix A, (Accessed on 10 April 2024, Link: https://urlz.fr/jlT1) and LFP recording (WT: 4.8 ± 1.2 vs. *scn1Lab*-KD: 28.2 ± 2.4, *p* < 0.0001) (Figure 4C,E_1_). More importantly, in microglia-depleted *scn1Lab*-KD larvae, an increased number of epileptiform events was observed when compared to that seen in *scn1Lab*-KD individuals with microglia. Moreover, the amplitude of the events detected in *scn1Lab*-KD larvae was also significantly increased in *scn1Lab*-KD individuals devoid of microglia (Figure 4D_2_,E_2_; Appendix A, (Accessed on 10 April 2024, Link: https://urlz.fr/jlTa, and Appendix A, (Accessed on 10 April 2024, Link: https://urlz.fr/jlTr). These data suggest that microglial cells in the *scn1Lab*-KD mutant context displayed neuroprotective activities that were able to mitigate, at least partially, the hyperactivation of neuron networks.

### 3.4. Microglia Ablation Exacerbates Seizure-like Swimming Behavior of scn1Lab-KD Larvae

*Scn1Lab* mutants and morphants exhibit spontaneous seizure-like swimming behavior characterized by whole-body convulsions, “swirling behavior”, and rapid undirected movements [31,38]. Therefore, we used the increased swimming behavior of *scn1Lab*-KD larvae to further question whether the increased neuron activity in *scn1Lab*-KD individuals could be modulated by the presence or absence of microglia (Figure 5A–C). As previously shown, *scn1Lab*-KD larvae swam longer and over longer distances than WT individuals (WT: 212 ± 11 vs. *scn1Lab*-KD: 355 ± 11 mm, *p* < 0.0001), reflecting their neuronal hyperactivation (Figure 5B–D_4_). Interestingly, in *scn1Lab*-KD larvae devoid of microglia, we observed a markedly increased locomotor activity compared to that observed in *scn1Lab*-KD larvae not microglia-depleted (*scn1Lab*-KD: 355 ± 11 vs. microglia-depleted *scn1Lab*-KD: 423 ± 12 mm, *p* < 0.0001) (Figure 5B–D_4_), suggesting that microglia depletion worsened the increased locomotor behavior of *scn1Lab*-KD larvae, and thus confirming that neuronal hyperactivation was further exacerbated in *scn1Lab*-KD larvae lacking microglia when compared to that observed in their siblings not microglia-depleted.

## 4. Discussion

It is well established that epileptic seizures induce microglia-mediated brain inflammation [48]. However, the precise response of microglia to epileptic seizures and its consequences on subsequent neuronal network function and activity have been poorly studied so far. In particular, it is not known whether seizure-induced microglia activities reduce or exacerbate neuronal network hyperactivity [12]. This gap in knowledge has several causes. Firstly, microglial cells cannot be directly imaged in vivo in rodents and most other animal species because of the opacity of their skull, necessitating transparent windows to be inserted through delicate surgery, thus increasing the risk of excessive and not-relevant differentiation of microglia, which are highly sensitive to disturbances in brain homeostasis [45]. Therefore, zebrafish larvae, with the optical transparency of their skulls, offer a powerful model that enables easy real-time imaging of microglial cells in live brains and, thus, in their physiological environment [43]. Secondly, most studies investigating microglia activities in rodent epilepsy models have been conducted using pharmacologically induced seizures. While those chemicals provided highly efficient models to promote neuron hyperactivity, they usually caused extended *status epilepticus*-like activity lasting for several hours [12,14,20,49], likely inducing much stronger brain inflammation than that observed in genetic models and probably in human patients [21,47,48]. This concern can be overcome by using genetic epilepsy models, such as those offered by zebrafish. In this study, using the zebrafish *scn1Lab*-KD DS model, we report here what is, to our knowledge, the first study analyzing the response of microglial cells in vivo in a DS model.

DS is a severe DEE induced by deficits in the activity of inhibitory interneurons and most commonly caused by de novo loss-of-function mutations in the *SCN1A* gene encoding the voltage-gated sodium channel NaV1.1. Models of this disease have been characterized in zebrafish using morpholino-mediated or loss-of-function mutations of one of the two zebrafish orthologues of the *SCN1A* gene, *scn1Lab*. Mutant and morphant *scn1Lab* larvae display recurrent epileptiform seizures from 4 dpf onward, which we and others have previously characterized [31,38]. In particular, we previously showed that loss of *scn1Lab* function induced a shift in the excitatory/inhibitory synapse balance, with a decreased number of GABAergic synapses and an increased number of glutamatergic ones, combined with an increased number of apoptotic neurons [31].

In close agreement with data gained from both animal models and human patients [50], microglia displayed significant morphological differences in *scn1Lab*-KD larvae when compared to those in their not-injected siblings. In the mutant context, we observed a decreased number of “branched” microglia and an increased percentage of cells showing an “amoeboid” morphology, the latter resembling M1-type pro-inflammatory “activated” microglia observed following brain injuries or in various neuronal disease situations, including epilepsy [18], but also, as we previously showed, following intoxication by neurotoxic organophosphates [37,51]. Interestingly, although DS larvae showed an increased number of microglia with an “activated” phenotype, the proportion of “transitional” microglia was similar in the two genetic contexts, and a small, but significant, percentage (10%) of “branched” (resting) microglia was still observed in *scn1Lab*-KD individuals. These observations highlight the complexity of the microglial response in epilepsy and, more importantly, are evidence that different populations of microglial cells are present in epileptic brains, each likely playing a distinct role in the pathophysiology of the disease. Although increased expression of pro-inflammatory mediators has been observed both in patients and in various animal epilepsy models [18,52], other studies have shown that anti-inflammatory cytokines are also increased in epileptic brains [28].

Surprisingly, and in contrast with the results gained in many pharmacological rodent epilepsy models, qRT-PCR analysis found no significant overexpression of pro-inflammatory or anti-inflammatory cytokine-encoding genes in the brains of *scn1Lab*-KD individuals. This result could be due to the moderate severity of the seizures observed in this context compared to those induced by pharmacological agents such as PTZ, kainate, or DFP [37]. However, using the transgenic Tg[il1β:GFP] line, which enables Il1β-expressing microglial cells to be visualized in vivo [35], we observed a significant increase in the percentage of cells expressing this pro-inflammatory cytokine in *scn1Lab*-KD individuals. This observation, which is in close agreement with our cluster analysis, further confirmed not only the increased number of M1-like “activated” microglia in *scn1Lab-KD* larvae but also the existence of at least two distinct microglia populations, expressing, or not, Il1β. However, this increased *il1β* expression was probably too weak to be detected by qRT-PCR analysis of brain RNAs, showing again the differences between genetic and pharmacological epilepsy models [37]. Interestingly, only part of the microglia expressed the Tg[il1β:GFP] transgene in *scn1Lab-KD* larvae, and while most of these cells displayed an “amoeboid” morphology, a few highly ”branched” cells also expressed the transgene. These results further demonstrate the complexity of the response of microglia to epileptic seizures. Moreover, these data also suggest that biochemical and morphological changes following microglia activation are not fully coupled, with cells displaying either or both features. It is also worth noting that studies of microglia-mediated neuroinflammation in *SCN1A*^+/−^ mice yielded widely varying results. In some studies, significant activation of microglial cells was observed in the prefrontal cortex and dentate gyrus, combined with an increased expression of the *IL1β* and *TNFα* genes [44], while in other studies, no microglia activation was detected in the hippocampus of *SCN1A*^+/−^ mice, associated with a lack of over-expression of pro-inflammatory mediators [53]. Moreover, Benson et al. (2015) showed that whereas microglia expressed both pro- and anti-inflammatory mediators in the mouse pilocarpine model, only pro-inflammatory mediators were overexpressed in microglia of mice exposed to kainate [28], highlighting again the wide range of microglia responses in epilepsy.

Genetic depletion of microglial cells in *scn1Lab*-KD larvae exacerbated the hyper-activity of neuronal networks, as shown by the increased number of spontaneous epileptiform seizures and worsening of the characteristic swirling swimming behavior of larvae, further suggesting that microglia played a protective role in epileptic brains and mitigated neuronal hyper-activity. This result is in good agreement with several studies, which concluded that microglia played a beneficial role in epilepsy through mitigation of neuronal activity and modulation of synaptic plasticity [25,26,53]. Also, Mirrione et al. (2010) showed that microglia ablation in adult mice led to an increased sensitivity to the pro-convulsant substance pilocarpine [54], further suggesting that microglia display neuroprotective functions in epileptic contexts. However, preventing microglia proliferation during the late stage of epileptogenesis, but not the early stage, decreased spontaneous seizures in the kainate mouse model, showing the complexity of the temporal responses of these cells [55]. It can be hypothesized that the neuroprotective activities of microglia involve the expression of anti-inflammatory cytokine and neurotrophic factors, such as brain-derived neurotrophic factor (BDNF). Microglia neuroprotection may also rely on the expression of protective H_2_O_2_ signaling to neurons or astrocytes. Further work will be necessary to better understand the molecular basis of the neuroprotective activities of these cells in epilepsy.

In the present work, which is, to our knowledge, the first study investigating the responses of microglial cells in vivo in a genetic DS epilepsy model, we showed that epileptic seizures induce deep microglia reprogramming, including significant changes in cell shape and dynamic behavior and increased expression of the pro-inflammatory cytokine Il-1β. However, this reprogramming is partial, and only part of the microglia shows either M1-type morphology or *il1β* expression, or both, suggesting that the response of these cells to seizures is complex and not limited to M1-type activation. Importantly, we also showed that *scn1Lab*-KD larvae entirely devoid of microglia exhibited an increased number of seizures, also displaying a greater amplitude, suggesting that microglia provide neuroprotection by decreasing the excitability of epileptic neurons. Future research will be necessary to investigate how microglial cells modulate neuronal activity, because modulation of the activity of these cells may offer a novel therapeutic strategy with great potential to mitigate neuron hyperactivity in epilepsy.

## Figures and Tables

**Figure 1 cells-13-00684-f001:**
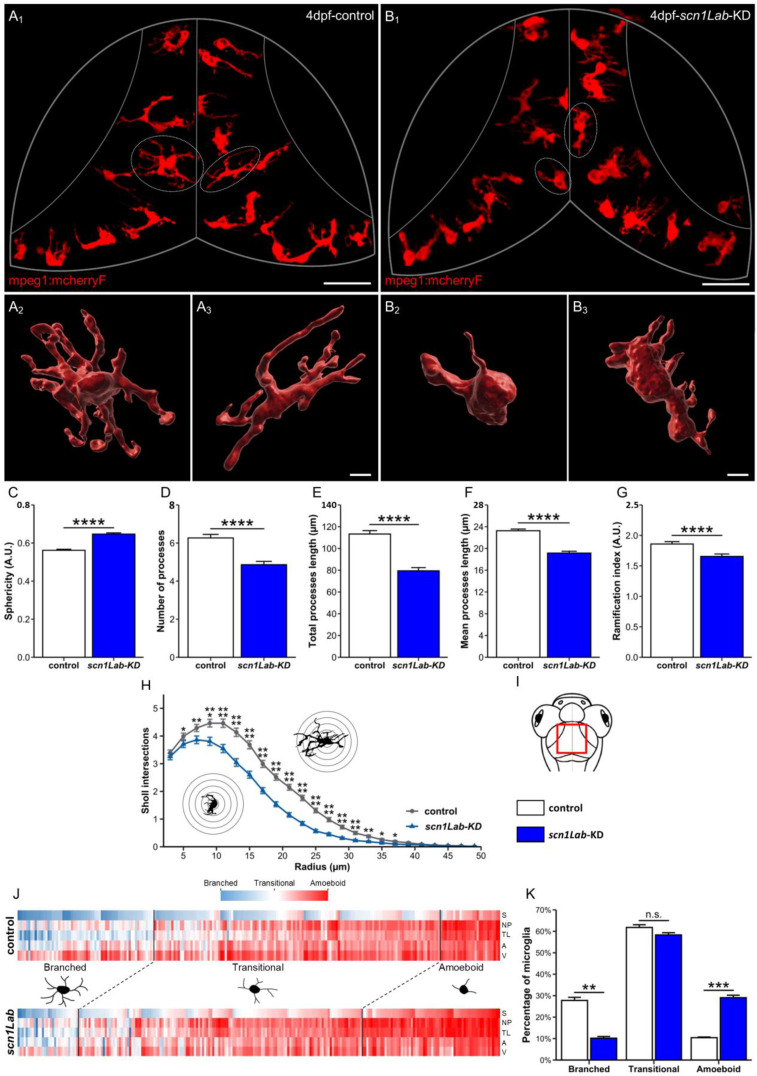
Microglia morphology parameters and clustering, in *scn1Lab*-KD larvae. (**A**,**B**) Dorsal views of the optic tectum of 4 dpf Tg[mpeg1:mCherryF] (**A_1_**) and Tg[mpeg1:mCherryF]; *scn1Lab*-KD (**B_2_**) larvae showing microglial cells, and 3D reconstructions of microglia in Tg[mpeg1:mCherryF] (**A_2_**,**A_3_**) and Tg[mpeg1:mCherryF]; *scn1Lab*-KD (**B_2_**,**B_3_**) individuals. (**C**–**G**) Quantification of microglia sphericity (**C**), number of processes (**D**), total process length (**E**), mean process length (**F**), and ramification index (**G**), in 4 dpf Tg[mpeg1:mCherryF]; *scn1Lab*-KD (N = 11; *n* = 239) and Tg[mpeg1:mCherryF] larvae (N = 11; *n* = 228). (**H**) Sholl analysis of Tg[mpeg1:mCherryF] (black) and Tg[mpeg1:mCherryF]; *scn1Lab*-KD (blue) microglia branching. (**I**) Scheme of the head of a zebrafish larva with the area of interest (the periventricular stratum) framed in red. (**J**) Cluster analysis of microglia populations in 4 dpf Tg[mpeg1:mCherryF]; *scn1Lab*-KD larvae (N = 11; n = 239) and Tg[mpeg1:mCherryF] sibling controls (N = 11; *n* = 228), according to sphericity (S), number of processes (NP), total process length (TL), area (A), and volume (V), leading to three cell populations. (**K**) Repartition of control (white) and *scn1Lab*-KD (blue) microglia populations in the three clusters defined (“branched”, “amoeboid”, and “intermediate”). All data are represented as the mean ± sem. *p*-values were determined using Mann–Whitney or unpaired Student *t*-test depending on the normal distribution of the values. n.s., non-significant; *, *p* < 0.05; **, *p* < 0.01; ***, *p* < 0.001; ****, *p* < 0.0001. Scale bars: 50 µm (**A_1_**,**B_1_**); 10 µm (**A_2_**,**A_3_**,**B_2_**,**B_3_**).

**Figure 2 cells-13-00684-f002:**
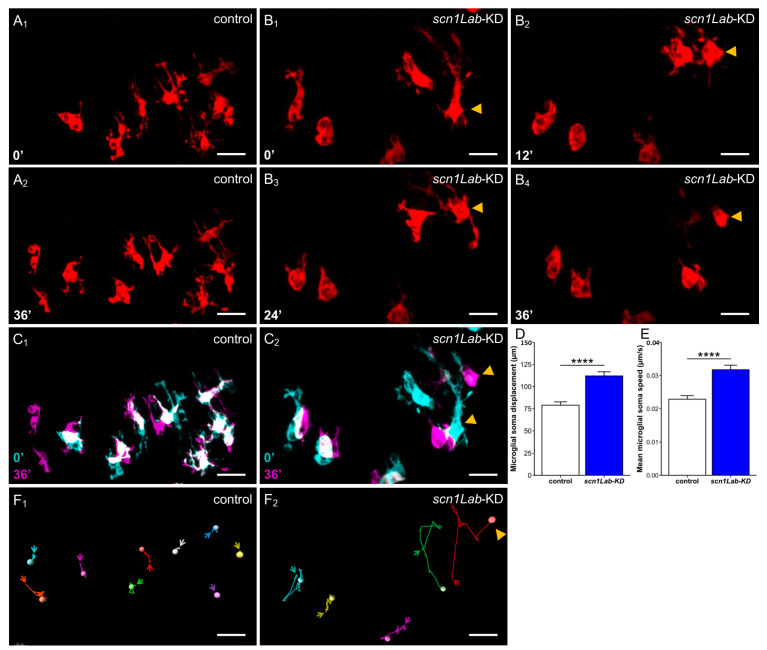
Microglia dynamics in *scn1Lab*-KD larvae. Confocal time-lapse analysis of microglia dynamics in the brains of 4 dpf not-injected Tg[mpeg1:mCherryF] control (**A_1_**,**A_2_**) and Tg[mpeg1:mCherryF]; *scn1Lab*-KD larvae (**B_1_**–**B_4_**), and merged images of two-time points highlighting the increased move of microglia cell bodies in *scn1Lab*-KD larvae (**C_2_**) compared to that observed in WT controls during a 36-min period (**C_1_**) (0′: cyan, 36′: magenta, merge: white). Quantification of microglia cell body displacements (**D**) and mean displacement speed (**E**) over a 60-min period, in 4 dpf *scn1Lab*-KD larvae (N = 6; *n* = 55) and WT controls (N = 4; *n* = 46). (**F**) Microglial cell tracking over a 60-min period, in 4 dpf control (**F_1_**) and *scn1Lab*-KD (**F_2_**) larva brain. The scale bar is 20 µm in all images. N = number of larvae and n = number of microglial cells. All images were acquired with an SP8 Leica laser scanning confocal equipped with a 40×/water 1.1 objective. All data are represented as the mean ± sem. *p*-values were determined using the Mann–Whitney test. ****, *p* < 0.0001.

**Figure 3 cells-13-00684-f003:**
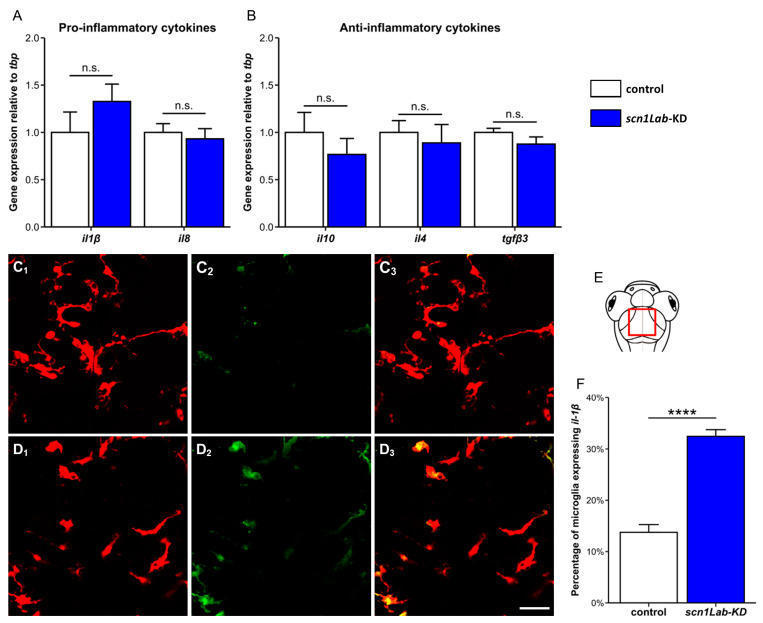
Cytokine expression in *scn1Lab*-KD larvae. (**A**,**B**) Quantification of the expression of genes encoding pro- (Il1β and Il8) (**A**), and anti-inflammatory cytokines (Il10, Il4, and Tgfβ3) (**B**), relative to that of the *tbp* gene, in the brains of 4 dpf not-injected (white) and *scn1Lab*-KD (blue) larvae. (**C**,**D**) Dorsal view of the periventricular stratum of 4 dpf Tg[mpeg1:mCherryF] (**C_1_**–**C_3_**) and Tg[mpeg1:mCherryF]; *scn1Lab*-KD larvae (**D_1_**–**D_3_**), showing Il1β-expressing microglial cells (**C_2_**,**D_2_**). Note that while part of “amoeboid” microglial cells only expressed the *il1β:GFP* transgene, few highly branched microglia also expressed *il1β:GFP*. (**E**) Scheme of a zebrafish larval head with the area of interest (the periventricular stratum) framed in red. (**F**) Quantification of the percentage of microglial cells expressing Il1β:GFP in not-injected (N = 19) and *scn1Lab*-KD larvae (N = 16). Scale: 10 µm. N = number of larvae analyzed. All images were acquired using a Leica SP8 confocal microscope, equipped with a 20×/water objective with a numerical aperture of 0.75. All graphs represent mean ± sem. The *p*-values were calculated using a Student’s *t*-test. n.s., not significant; ****, *p* < 0.0001.

**Figure 4 cells-13-00684-f004:**
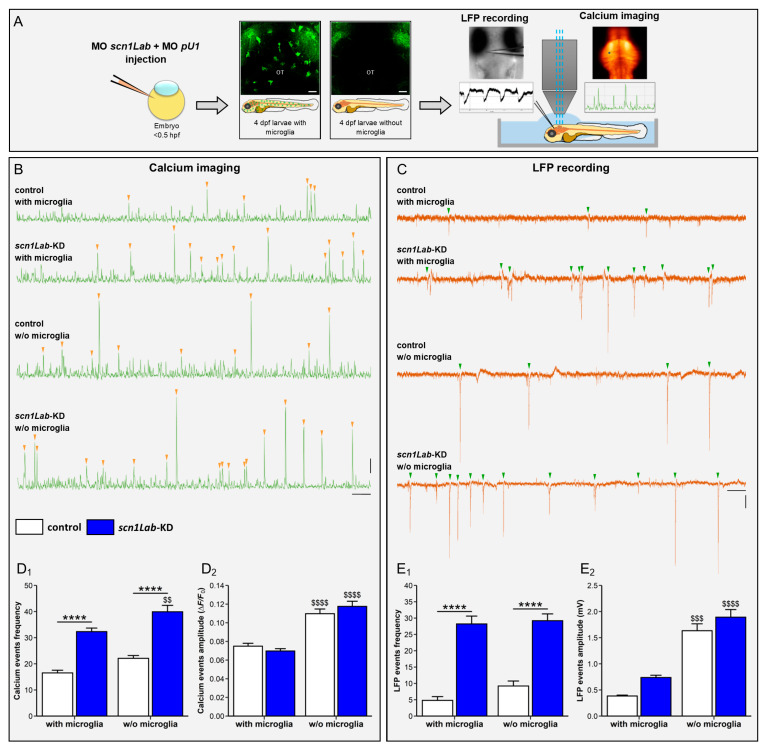
Microglia depletion increases neuron hyperactivity in *scn1Lab*-KD larvae. (**A**) Scheme of the experimental setup. (**B**) Calcium imaging of Ca^++^ uptake events recorded over a 20-min period in 4 dpf WT (N = 11), *scn1Lab*-KD (N = 16), microglia-depleted WT (N = 9) and microglia-depleted *scn1Lab*-KD larvae (N = 9). (**C**) LFP recording over a 20-min period of neuron activity in 4 dpf WT (N = 5), *scn1Lab*-KD (N = 5), microglia-depleted WT (N = 5) and microglia-depleted *scn1Lab*-KD larvae (N = 5). Scale bars of calcium traces: 0.05 ΔF/F_0_ and 1 min; and that of LFP traces: 0.5 mV and 1 min. Quantification of the frequency (**D_1_**) and amplitude (**D_2_**) of calcium events with intensity > 0.04 ΔF/F_0_, recorded during a 1-h period in 4 dpf not-injected (white) and *scn1Lab*-KD larvae (blue). (**E**). Quantification of the frequency (**E_1_**) and amplitude (**E_2_**) of LFP traces with downwards depolarization > 0.3 mV and lasting > 100 ms, recorded during a 1-h period in 4 dpf not injected (white) and *scn1Lab*-KD larvae (blue). N = number of larvae. All data are represented as the mean ± sem. ****, *p* < 0.0001: indicates a statistically significant difference between control and *scn1Lab*-KD larvae. $$, *p* < 0.01; $$$, *p* < 0.001; $$$$, *p* < 0.0001: indicates a statistically significant difference between larvae with or without microglia. *p*-values were determined using two-way ANOVA with Tukey post-tests.

**Figure 5 cells-13-00684-f005:**
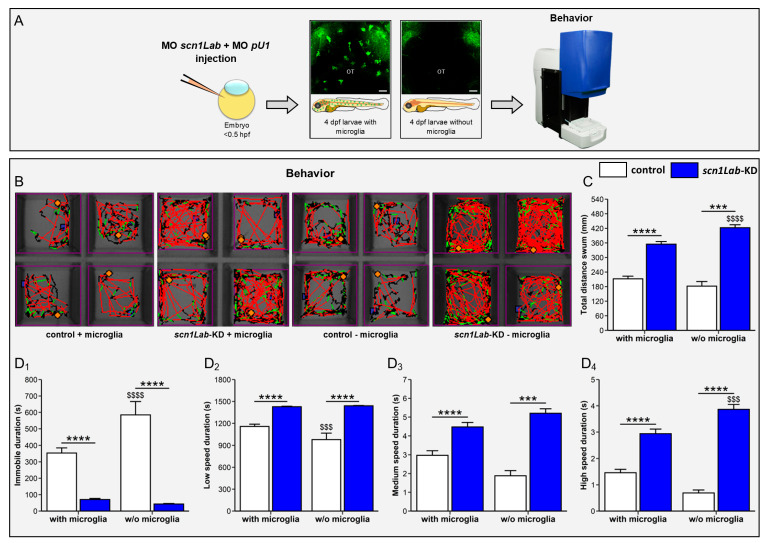
Microglia ablation increases the locomotor activity of *scn1Lab*-KD larvae. (**A**) Scheme of the experimental setup. (**B**) Traces of the moves of WT (N = 148), *scn1Lab*-KD (N = 184), microglia-depleted WT (N = 45), and microglia-depleted *scn1Lab*-KD larvae (N = 28). The swimming speeds of larvae are represented according to the following color codes: black (low speed), green (medium speed), and red (high speed). (**C**) Quantification of the total distance traveled by 4 dpf WT (white) and *scn1Lab*-KD larvae (blue) during a 30-min period recording. (**D**) Quantification of the time spent by 4 dpf WT (white) and *scn1Lab*-KD larvae (blue) in a motionless position (**D_1_**), or swimming at low (**D_2_**), medium (**D_3_**), and high speed (**D_4_**). N = number of larvae. All graphs represent the average ± sem. The *p*-values were calculated by a two-way ANOVA followed by a Tukey post-test. ***, *p* < 0.001; ****, *p* < 0.0001: indicates a statistical difference between genotypes. $$$, *p* < 0.001; $$$$, *p* < 0.0001: indicate significant differences depending on the presence or absence of microglia.

## Data Availability

All data are presented in the article.

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
