# Peer review of "Microglia Mitigate Neuronal Activation in a Zebrafish Model of Dravet Syndrome"

_cells, 2024, doi:10.3390/cells13080684_

Round 1
Reviewer 1 Report
Comments and Suggestions for Authors
For quite some time, the understanding has been that epileptic seizures trigger neuroinflammation in the brain by activating microglia. However, the importance of these cells in epilepsy studies has been overlooked, and the impact of their response on subsequent brain function is not well comprehended. In this research, we aimed to bridge this knowledge gap and enhance our comprehension of microglia's significance in epilepsy's pathogenesis, employing a well-established zebrafish model of Dravet syndrome. This model is characterized by the dysfunction of the Scn1Lab sodium channel, alongside techniques such as live imaging of microglia and neuronal Ca2+ levels, recording of local field potentials (LFP), and genetic elimination of microglia. Initially, we observed that larvae deficient in scn1Lab, undergoing epileptic seizures, demonstrated microglia with physical and chemical signs of M1-like pro-inflammatory activation. This included a decrease in branching, a change to an amoeboid form, and a notable rise in microglia expressing the pro-inflammatory cytokine Il1β. More critically, larvae with scn1Lab knockdown but completely devoid of microglia exhibited significantly higher neuronal activity compared to those with intact microglia, as evidenced by LFP recordings and Ca2+ imaging, as well as by the seizure-induced erratic swimming behavior of the larvae. These results indicate that although microglia become activated and produce pro-inflammatory cytokines, they play a neuroprotective role in epileptic neural circuits, highlighting their potential as a therapeutic target for epilepsy. I find it an excellent experimental approach to a disease.
Comments on the Quality of English LanguageEnglish language, acceptable
Author Response
We would like to thank this reviewer for his attention to our manuscript and for his comments.
Reviewer 2 Report
Comments and Suggestions for Authors
The authors investigated a role for microglia in epilepsy using a zebrafish model of Dravet syndrome, where a neuronal sodium channel is mutant or knocked down using morpholino technology. They found that microglia in this epilepsy syndrome model transitioned to a shape more characteristic of the M1 proinflammatory state. Even though direct measurements of brain proinflammatory cytokines did not show an increase in the Dravet syndrome model, experiments with GFP reporter strains for proinflammatory cytokine expression showed an increase in microglial expression that was not detected by qPCR of the entire brain. Fluorescent measurements of neuronal cytoplasmic Ca2+ showed neuronal hyperactivation in the Dravet syndrome fish. Perhaps the most important results come from experiments of genetic microglial ablation showing that overall microglia are protective in this seizure model. These experiments take great advantage of the in vivo imaging capabilities of the transparent fish model system.
Major comments: The experiments were well-planned and carried out. The links to the data videos were helpful. The data analysis was appropriate and the manuscript is well-organized with sufficient background information and the results are laid out clearly with a balanced discussion of the results in the context of precious data. Therefore, there are no major negative comments.
Minor comments:
Comment 1: Line 447 contains a sentence fragment (no verb). Please combine with the previous sentence by replacing the period with a comma: [59]. All -> [59], all
Comment 2: In the Discussion please speculate on how microglia could be protective against epilepsy. Is it likely through anti-inflammatory cytokine expression or perhaps through protective H2O2 signaling to neurons or astrocytes by NADPH oxidases that induces hormesis and the activation of stress responses, such as Nrf2 in astrocytes?
Author Response
We would like to thank this reviewer for his comments on our manuscript and his helpful suggestions.
As a reply to the minor comments addressed:
- In the line 447, the sentence fragment without verb was eliminated during the deep revision of the text of our manuscript.
- In the Discussion section, as suggested by this reviewer, we speculated on how microglia could be protective in epilepsy, hypothetically through anti-inflammatory cytokine expression or protective H2O2 signaling to neurons or astrocytes by NADPH oxidases.
Reviewer 3 Report
Comments and Suggestions for Authors
The present study tried to gain a larger understanding of the role of microglia in the pathophysiology of epilepsy, using an established zebrafish Dravet syndrome epilepsy model based on Scn1Lab sodium channel loss-of-function, combined with live microglia and neuronal Ca2+ imaging, local field potential (LFP) recording and genetic microglial ablation. The results showed that in scn1Lab-deficient larvae experiencing epileptiform seizures, microglia displayed morphological and biochemical features suggesting M1-like pro-inflammatory activation, including reduced branching, amoeboid-like morphology, and a marked increase in the number of microglia expressing pro-inflammatory cytokine interleukin-1β. More importantly, scn1Lab-KD larvae fully lacking microglia showed a significantly increased neuronal activation compared to that seen in scn1Lab-KD 22 individuals with microglia, as shown by LFP recording and Ca2+ imaging, and also by the epileptiform seizure-related whirling swimming of larvae. These findings were evidence that despite a microglial activation and the synthesis of pro-inflammatory cytokines, microglia also provide neuroprotection to epileptic neuronal networks.
There are some concerns as listed in the following, especially the inconsistent writing format for References:
*L152: 40 minutes; L179: 30 minutes -> minutes vs. min
**L377: (Figures 5F, 5G, 5H). -> not appeared in the Figure 5
***Inconsistent writing format for References
L543: R12: no page number
L563: R20: Journal of Neuroscience -> Abbreviation
L573: R23: Journal of Neuroscience -> Abbreviation
L577: R25: no page number
L585: R28: no page number
L591: R31: no page number
L595: Am. J. Hum. Genet
L598: R34: Journal of Neuroscience -> Abbreviation
L601: R35: Journal of Neuroscience -> Abbreviation
L607: R37: no page number
L610: R38: no page number
L616: R41 is repeated with R40
L633: R47: no page number
L637: R49: no page number
L642: R51: no page number
L645: R52: no page number
L648: R53: no page number
L651: R54: no page number
L654: R55: no page number
L658: R57: no page number
Author Response
We would like to thank this referee for his comments and for careful reading of our article.
To reply to his minor comments:
- We have standardized the writing of min/minutes throughout the text.
- In lane 377, we have modified Figures 5F, 5G, 5H to Figures 5D1-D4, in agreement with the panels shown in Figure 5.
- In lane 543, page number was added.
- In lane 563, Journal of Neuroscience -> J. Neurosci.
- In lane 573, Journal of Neuroscience -> J. Neurosci.
- In lane 577, page number was not added because BMC publications do not have page number.
- In lane 585, page number was not added because BMC publications do not have page number.
- In lane 591, page number was added.
- In lane 595: Am. J. Hum. Genet -> Am. J. Hum. Genet.
- In lane 598, Journal of Neuroscience -> J. Neurosci.
- In lane 601, Journal of Neuroscience -> J. Neurosci.
- In lane 607, page number was not added because Elife articles do not have page number.
- In lane 610, page number was not added because Frontiers articles do not have page number.
- In lane 616: R41, a repetition with R40, was eliminated.
- In lane 633, page number was not added because Elife articles do not have page number.
- In lane 637, page number was not added because eNeuro articles do not have page number.
- In lane 642, page number was not added because BMC publications do not have page number.
- In lane 645, page number was not added because Neurobiol. Dis. articles have no page number.
- In lane 648, R53 was removed because this reference is repeated with R52.
- In lane 651, page number was not added because MDPI articles do not have page number.
- In lane 654, page number was not added because Neuropathol. Appl. Neurobiol articles do not have page number.
- In lane 658, page number was not added because MDPI articles do not have page number.